# Experience Replay More When It's a Key Transition in Deep Reinforcement Learning

## Abstract

We proposed an experience replay mechanism in Deep Reinforcement Learning based on Add Noise to Noise (AN2N), which requires agent to replay more experience containing key state, abbreviated as Experience Replay More (ERM). In the AN2N algorithm, we refer to the states explored more as the key states. We found that how the transitions containing the key state participates in updating the policy and Q networks has a significant impact on the performance improvement of the deep reinforcement learning agent, and the problem of catastrophic forgetting in neural networks is further magnified in the AN2N algorithm. Therefore, We sample the transition used for experience replay according to whether the transition contains key states and whether it is the most recently generated, rather than sampling uniformly, which is the core idea of the ERM algorithm. The experimental results show that this algorithm improves the performance of the agent by a wide margin. We combine the ERM algorithm with Deep Deterministic Policy Gradient (DDPG), Twin Delayed Deep Deterministic policy gradient (TD3) and Soft Actor-Critic (SAC), and evaluate algorithm on the suite of OpenAI gym tasks, SAC with ERM achieves a new state of the art, and DDPG with ERM can even exceed the average performance of SAC under certain random seeds, which is incredible.

## 1 Introduction

Deep reinforcement learning (RL) has shown its promising future for decision-making in various computer games, such as atari (Mnih et al., 2013; 2015), go (Schrittwieser et al., 2020) and starcraft (Vinyals et al., 2019). However, most successes have been exclusively in simulation due to poor sample efficiency of typical Deep RL algorithm and other challenges. Reinforcement learning can be divided into model-based RL and model-free RL in the light of its data efficiency. Model-free RL is usually subdivided into off-policy RL and on-policy RL. Although model-based RL requires less sampled data, it needs to build a world model and predict the next state based on a lot of prior work, demonstrated in Hafner et al. (2019). Although the on-policy RL algorithm do without establishing a world model and has good stability, the performance improves slowly due to limiting the step of updating the policy, a large number of sampled trajectory data are required in the training process (Schulman et al., 2015; 2017). Off-policy RL is between model-based RL and on-policy RL in terms of sampling efficiency, and the research in this field is enduring (Watkins & Dayan, 1992; Hessel et al., 2018; Barth-Maron et al., 2018) on account of its relatively high data efficiency and world model free.

Experience Replay (Lin, 1992) is an important part of improving the data efficiency of Off-policy RL, which stores experience in a replay buffer and break the temporal correlations by mixing data, therefore, the experience can be used multiple times to update the networks. However, most of the current work is to uniformly sample transitions from the buffer, such as Deep Deterministic Policy Gradient (DDPG) (Lillicrap et al., 2015), Soft Actor-Critic (SAC) (Haarnoja et al., 2018), Twin Delayed Deep Deterministic policy gradient (TD3) (Fujimoto et al., 2018) and many other algorithms (Van Hasselt et al., 2016; Mnih et al., 2016; Andrychowicz et al., 2017; Dabney et al., 2018; Liu et al., 2020). However, these approaches

replay experience transitions at the same frequency, regardless of their meaning. Schaul et al. (2016) develops a framework for prioritizing experience to replay important transitions more frequently, therefore agent learns more efficiently. Prioritized experience replay (PER) method samples transitions with the magnitude of their temporal-difference (TD) error. However, prioritization introduces bias, which needs to be corrected with importance sampling. Meta-reinforcement learning (meta-RL) algorithms enable agents to learn new skills from small amounts of experience (Rothfuss et al., 2018; Mishra et al., 2018), Rakelly et al. (2019) develops an off-policy meta-RL algorithm that disentangles the inference and control of the task, but its performance is still far behind the mainstream off-policy RL algorithms.

In this paper, We first analyzed how the state changes from beginning of agent interacting with the environment to the policy converged, and found that the state of the agent is different at different stages, agent will rarely transfer to those states where the policy is far from converging when the agent's policy already converged. If a large number of initial-states[1] are used to train Q-value and policy networks, the network weights that have been learned well will be gradually updated. Therefore, the sampling proportion of the most recently generated transitions should be appropriately increased, so that the agent pays more attention to learning the recent transitions, In this way, the state of the agent is gradually transferred from a poor state to a better state, similar to updating from a poor policy to a better policy stably in Trust region policy optimization (TRPO) (Schulman et al., 2015) or Proximal policy optimization (PPO) (Schulman et al., 2017).

Inspired by the Add Noise to Noise (AN2N) Algorithm (Guo & Gao, 2021), We divide the states into two categories. These states explored with added noise are called key states[2], and the rest are called non-key states. For the sake of improving the performance in key states in time, we have increased the probability of sampling new key states, making them more likely to participate in updating policy, we call this process as ERM (Experience Replay More), and combine it with commonly used off-policy RL algorithms in continuous control tasks, such as SAC, which obtained faster convergence and state-of-the-art performance.

## 2 Preliminaries

We consider a reinforcement learning setup consisting of agent learning policies to maximize the expected reward when interacting with the environment (Sutton & Barto, 2018). At each timestep $t$, the agent receives an observation $o_t \in \mathcal{O}$, selects action $a_t \in \mathcal{A}$ with respect to its policy $\pi: \mathcal{O} \to \mathcal{A}$. After taking the action $a_t$ in environment $E$, agent receives a reward $r_t$ and the next observation $o_{t+1}$. The practical problem is usually a partial Markov decision process (POMDP), only part of the observation information could be obtained. To simplify the problem, we assumed the environment is fully-observed, so $s_t = o_t, \mathcal{S} = \mathcal{O}$.

In reinforcement learning, the action-value function $Q^\pi(s, a)$ is uesed to approximate the expected sum reward of the ation $a$ in state $s$, defined as following:

$$Q^\pi(s, a) = \mathbb{E}_{s_t \sim p_\pi, a_t \sim \pi} \left[ \sum_{t=0}^{+\infty} \gamma^t R(s_t, a_t) \right] \tag{1}$$

Where $\gamma \in [0, 1]$ is the discount factor, $\mathbb{E}_{s_t \sim p_\pi, a_t \sim \pi}$ is the expectation over the distribution of the trajectories $(s_0, a_0, s_1, a_1, \dots)$.

The mean value of $Q^\pi$ in the same state $s$ called the value function $V^\pi$, defined as $V^\pi(s) = \mathbb{E}_{a \sim \pi(\cdot|s)}[Q^\pi(s, a)]$. We express the action-value function $Q^\pi$ in the form of Bellman equation (Bellman & Kalaba, 1965):

$$Q^\pi(s_t, a_t) = \mathbb{E}_{s_{t+1} \sim p_\pi}[r(s_t, a_t) + \gamma \mathbb{E}_{a_{t+1} \sim \pi}[Q^\pi(s_{t+1}, a_{t+1})]] \tag{2}$$

---

[1]The initial-state is the state in which agent do not perform well when policy has not converged, agent almost no longer transferred to this kind of state after policy converged.

[2]The key state is a state where the agent has not performed well in the past

In this paper, we need to be familiar with DDPG, TD3 and SAC algorithms, here, we mainly introduce DDPG as the basis. DDPG applied two different fully connected neural networks to approximate the action-value function $Q(s, a|\theta^Q)$ and policy function $\mu(s|\theta^\mu)$, DDPG introduces action-value target network $\theta^{Q'}$ and policy target network $\theta^{\mu'}$, so as to Stable the policy update. Consequently, gradient descent is used to optimize the network weights by minimizing the loss:

$$L(\theta^Q) = \mathbb{E}_{s_t \sim p_{\mu(s_t|\theta^\mu)}, a_t \sim \mu(s_t|\theta^\mu)}[(Q(s_t, a_t|\theta^Q) - y_t)^2] \tag{3}$$

Where

$$y_t = r(s_t, a_t) + \gamma Q'(s_{t+1}, \mu'(s_{t+1}|\theta^{\mu'})|\theta^{Q'}) \tag{4}$$

$$\nabla_{\theta^\mu} J \approx \mathbb{E}_{s \sim p_{(s_t|\theta^\mu)}} \left[ \nabla_{\theta^\mu} Q(s, a|\theta^Q)|_{s=s_t, a=\mu(s_t|\theta^\mu)} \right]$$
$$= \mathbb{E}_{s \sim p_{(s_t|\theta^\mu)}} [\nabla_a Q(s, a|\theta^Q)|_{s=s_t, a=\mu(s_t)} \nabla_{\theta^\mu} \mu(s_t|\theta^\mu)|s = s_t] \tag{5}$$

Where equation 4 derived from equation 2, the weights of target networks are updated periodically to slowly track the learned networks: $\theta' \leftarrow \tau\theta + (1 - \tau)\theta'$ with $\tau \ll 1$, which alleviates the fluctuation in the agent's learning process. The policy is updated by equation 5, following the chain rule to the expected sum return $Q(s, a|\theta^Q)$ with respect to parameters $\theta^\mu$. TD3 addresses the problem that DDPG is prone to overestimating the Q function. An additional Q-function is added, and the predicted smaller Q value is used as the calculation of TD-error. At the same time, which also reduces the update frequency of the policy function. The most significant difference between SAC and DDPG is the introduction of policy entropy $H(\pi(\cdot|s_t))$, so the objective function equation 1 is rewritten as:

$$Q^\pi(s, a) = \mathbb{E}_{s_t \sim p_\pi, a_t \sim \pi} \left[ \sum_{t=0}^{+\infty} \gamma^t R(s_t, a_t) - \alpha \log(\pi(a_t|s_t)) \right] \tag{6}$$

Where $\alpha$ is temperature parameter, which adjusts the optimization target, agent pays more attention to exploration if increase the coefficient $\alpha$.

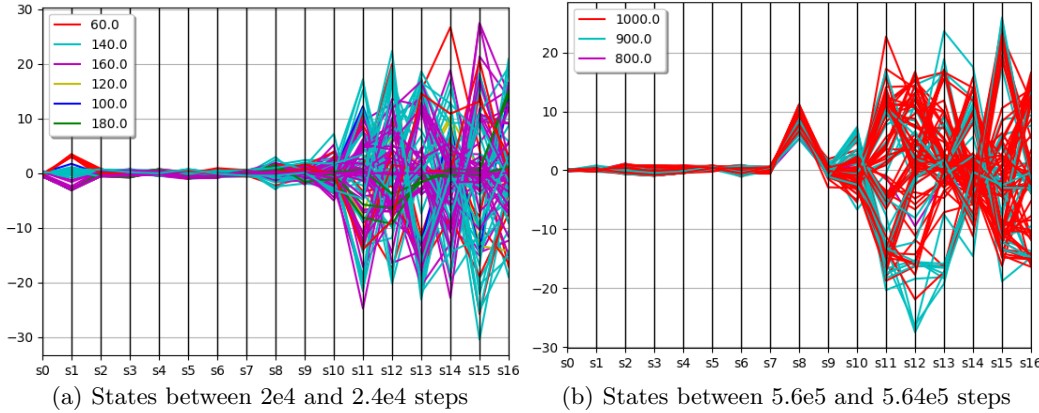

(a) States between 2e4 and 2.4e4 steps      (b) States between 5.6e5 and 5.64e5 steps

Figure 1: In the HalfCheetah-v2 environment, the state of the agent in different training stages is compared. The abscissa means the different positions on the agent, and the ordinate means the collected attitude information at the specific position, the difference is obvious at position s1, s8, s12 and s14. Different colors of legend indicate different Q values.(a) Start collecting at 2e4.(b) Start collecting at 5.6e5.

## 3   Experience Replay More When It's a Key Transition

The addition of experience replay improves the sample efficiency of off-policy RL. Many off-policy algorithms usually sample transitions uniformly from the experience buffer, which

potentially considers the transitions generated at different times to be of the same importance, we will discuss this in section 3.1. Prioritized Experience Replay (PER) is an optimization of the uniform sampling method based on the TD-error value of transitions so as to learn the samples more efficiently, however, PER essentially still does not consider whether the transitions generated at different times are equally important to the current agent's policy.

## 3.1 States are Different at Different Stages

Taking the HalfCheetah-v2 simulation environment as an example, we recorded the state information of an agent at different stages based on DDPG algorithm, collecting attitude information at the specific position and the corresponding Q-value starting from 2e4 to 5.6e5, and then we uniformly sample 100 sets of data from the collected data for drawing, as shown in the Fig. 1. The legend in the upper left corner represents the Q value, which is discretized to reasonably reduce the number of Legends. Subgraph (a) is discreted in units of 20, and subgraph b is discreted in units of 100. It can be found when the agent interacts with the environment, not only the policy is gradually improving[3], but the state is also changing accordingly.

After policy converges and the agent is in a state set different from the previous one, more experience similar to the agent's recent states should be collected to train the neural network to strengthen the current policy. If the sampling experience for training policy network and the action state network are quite different from the recently generated states, it will not only help improve the policy to a small extent, but will also even forget the newly learned network weights due to the catastrophic forgetting of the neural networks. Therefore, to further improving the sample efficiency, experience replay should be switched from uniform sampling to targeted sampling.

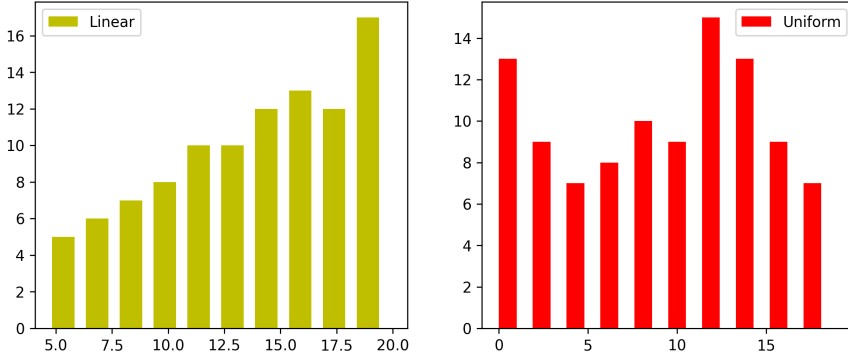

Figure 2: Compare the sampling results by using the linear distribution sampling function and the uniform distribution sampling function. The left picture shows the sampling result by linear distribution, and the right picture shows another. We sampled 100 samples in $[0, 20]$.

## 3.2 Replaying More Key Transitions

The AN2N algorithm will record the dilemma state $s_d$ during the evaluation process, so when agent interacts with the environment, if the current state $s_c$ and $s_d$ are considered to be highly similar, an additional exploration noise $\mathcal{N}_a$ will be added. In this paper, the state $s_c$ with additional noise is called the key state $s_k$, and the transition containing $s_k$ is called the key transition $tran_k$. The agent's policy is relatively fragile in $s_k$, if agent has not learned how to get out of this dilemma before, agent is more likely to fall into a series of bad

---

[3]Better decision-making ability usually gets a higher Q value

states after $s_k$, which greatly reduces the agent's overall performance. Therefore, whether the agent can learn a good policy in $s_k$ is very important for improving the performance.

It can be seen from equation 2 that reward of the key state $s_k$ will affect the Q value of the state at the previous moment through the iteration of the dynamic equation, nevertheless, if we uniformly sample experience for training the networks, there will be the following three problems:

- Since most of the experience generated by AN2N are non-key states, the probability of sampling key states $s_k$ in the experience buffer is small.
- The key state $s_k$ is time-sensitive. The new key state needs to participate in training the network as quickly as possible. Otherwise, if agent's state undergoes a relatively change, its role in improving the policy will be declined.
- Since the agent's state is gradually changing, more recent the sampled experience, more similar its distribution is to the distribution of the latest experience generated currently, and the more it satisfies the assumption of independent and identical distribution (iid).

In response to the first question, we designed two experience buffers to store key transition and non-key transition respectively, and used $\min(Prt_{AN2N}, K_t)$ to adjust the proportion of sampling key transition, where $Prt_{AN2N}$ is the proportion of the number of key transitions generated by AN2N, $K_t$ is linearly related to the simulation times of the agent. The above ensures that key transitions can be sampled strictly according to the proportion from the experience buffer. For the second and third questions, we will linearly increase the probability of new transition being sampled. Two sampling functions will sample 100 samples in $[0, 20]$ to compare the difference of the linear distribution sampling function and the uniform distribution more vividly, the result shown in Fig. 2. The pseudo code of ERM algorithm is shown in algorithm 1.

---

Algorithm 1: ERM

---

Input: Sampling ratio $Prt_{AN2N}$, $K_t$, Replay buffer $R_{non-key}$, $R_{key}$, batch size $bs_1$, $bs_2$, $bs_{sum}$ and AN2N parameters

Randomly initialize critic network $Q(s, a|\theta^Q)$ and actor $\mu(s|\theta^\mu)$ with weights $\theta^Q$ and $\theta^\mu$

Initialize target network $Q'$ and $\mu'$ with weights $\theta^{Q'} \leftarrow \theta^Q$, $\theta^{\mu'} \leftarrow \theta^\mu$

for episode $e \in \{1,...,M\}$ do
    Initialize a random process $\mathcal{N}$ for action exploration
    Receive initial observation state $s_1$
    for $t \in \{1,...,T\}$ do
        Execute AN2N action $a_t$ and observe reward $r_t$ and observe new state $s_{t+1}$
        if (AN2N exploring more) then
          |  Store key transitio $(s_t, a_t, r_t, s_{t+1})$ in $R_{key}$
        else
          |  Store transition $(s_t, a_t, r_t, s_{t+1})$ in $R_{non-key}$
        end
        $bs_1 = \min(Prt_{AN2N}, K_t)$
        $bs_2 = bs_{sum} - bs_1$
        Sample $bs_1$ and $bs_2$ transitions with linear distribution in $R_{key}$ and $R_{non-key}$ for training
        Run DDPG, SAC or TD3 etc. Algorithm
    end
end

---

## 4 Experiments

In this section, we test the performance of the combinations of Experience Replay More algorithm (ERM) with different off-policy RL algorithms acrossing a variety of continuous control tasks. As is shown in Fig. 3, consistent with the benchmark, we use the mainstream Mujoco physics engine (Todorov et al., 2012) as the simulation environment to test

the performance of the algorithm in the HalfCheetah-v2, Swimmer-v2, Walker2d-v2, and Hopper-v2 tasks, as Mujoco offers a unique combination of speed, accuracy and modeling power, and it is also the first full-featured simulator designed from the ground up for the purpose of motion control. For the specific introduction of the task environment is shown in Table 2 in Appendix A.

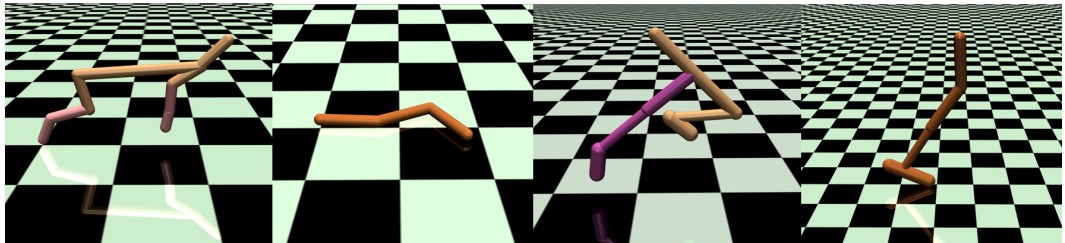

Figure 3: Samples of Mujoco tasks. In order from the left: HalfCheetah-v2, Swimmer-v2, Walker2d-v2, Hopper-v2.

We present the ERM by classifying key transitions and sampling them with linear distribution functions for the purpose of increasing the stability and performance with sampling efficiency, described in section 3. In each task, we run our algorithm and test it without exploration noise. The goal of our experimental evaluation is to understand how good the sample efficiency and stability of our method are, compared with prior off-policy RL algorithms. In all tasks, we run experiments for five times, which fix random seeds in 0, 5, 10, 15, 20 respectively. The results of ERM combined with different off policy RL algorithms are analyzed below.

### 4.1 DDPG with ERM

For the implementation of DDPG with ERM, we use a two layer fully connected network consists of $256 \times 256$ hidden nodes respectively, with rectified linear units (ReLU) between each layer for both the policy and action state networks, and a final tanh unit following the output of the policy network for limiting amplitude. After a certain number of steps, the networks are trained with a mini-batch of 100 transitions repeatedly, sampled from a replay buffer containing the entire history of the agent. See Appendix A for more experimental details.

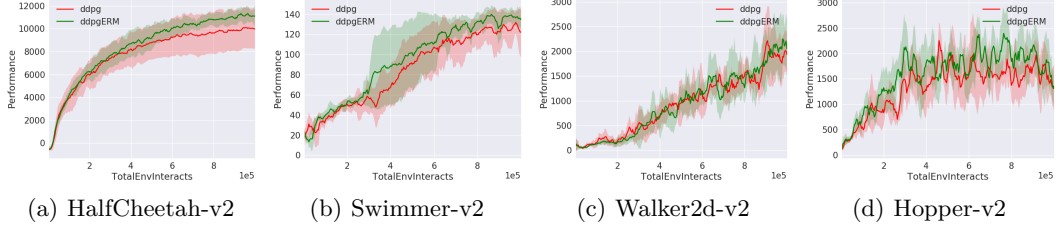

(a) HalfCheetah-v2  (b) Swimmer-v2  (c) Walker2d-v2  (d) Hopper-v2

Figure 4: Performance curves of the Agents using DDPG and DDPG with ERM: DDPG (red), DDPG with ERM (green).

DDPG with ERM uses two actor-networks and critic-networks respectively to approximate the policy and action-state value. In the test stage, we record the reward of each state of the agent and then calculate all the action state values of the trajectory when the episode finished, save those transitions whose total rewards are minimal. When the agent interacts with the environment, it uses the policy superimposed a small disturbance noise, if the current state is similar to key states, add a disturbance noise on small noise, otherwise, only use the small noise. Besides, we store the key tansitions and non-key transitions separately, which is convenient for sampling two kinds of transitions in a appropriate proportion with

linear distribution sampling function. The pseudo code of DDPG with ERM is shown in algorithm 2 in Appendix B.

We compare the DDPG with ERM algorithm with DDPG baseline, illustrated in Fig. 4, among the four tasks, the performance of DDPG with ERM is higher. In the HalfCheetah task, the method combined with ERM has significantly lower variance while maintaining higher performance, indicating that the algorithm we proposed with DDPG has better stability and sample efficiency.

## 4.2 TD3 with ERM

TD3 is an optimized version on the basis of DDPG, and the structure is very similar to the DDPG algorithm. The main three improvements are for the problems of DDPG in engineering practice: 1.Clipped Double-Q Learning. TD3 uses two Q networks with the same structure to predict the state action value at the same time, but when calculating TD-error, only the Q value with the smallest prediction participates in the calculation, so as to alleviate the overestimation of DDPG. 2.Delayed Policy Updates. Since updating the policy frequently is prone to make the policy unstable, therefore, TD3 updates the policy with a lower frequency. 3.Target Policy Smoothing. When calculating target Q, a small range of noise is added to the policy to make the calculated target Q value more robust. Therefore, TD3 with ERM can maintain the same network structure and hyper-parameters with DDPG with ERM. See Appendix A for more details.

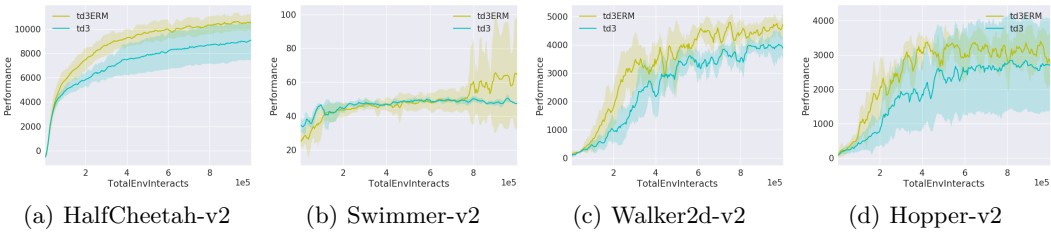

(a) HalfCheetah-v2  (b) Swimmer-v2  (c) Walker2d-v2  (d) Hopper-v2

Figure 5: Performance curves for a selection of domains using TD3 and TD3 with ERM: TD3 (cyan), TD3 with ERM (yellow).

In the same tasks with above, we tested the performance of TD3 and TD3 with ERM. As illustrated in Fig. 5. In all the test tasks, the performance improvement of TD3 with ERM is very obvious compared to TD3, especially in HalfCheetah and Walker2d tasks. At the same time, the method of combining TD3 and ERM has significantly lower variance while maintaining higher performance in the HalfCheetah and Hopper tasks, indicating that the algorithm we proposed is very stable and sample efficient on TD3.

## 4.3 SAC with ERM

Compared with TD3, SAC is more different from DDPG, but it is still an Actor-Critic structure. The main differences are as follows: 1.The policy entropy is added to the objective function, which can more fully explore the action space, but there is also an additional temperature coefficient that adjusts the entropy proportion of the policy. 2.SAC does not directly outputting the deterministic policy, but a mean value and variance of a policy's distribution, so it is necessary to adjust the range of the variance to achieve a noise-like addition. See Appendix A for the specific hyper-parameter settings of SAC.

The SAC algorithm is currently one of the most commonly used off-policy RL methods in academic and industry owing to its good performance, stability and sample efficiency. We tested the performance of SAC and SAC with ERM in four tasks, as shown in Fig. 6, we found that in half of the tasks, the performance of SAC with ERM still has a performance improvement compared to sac, while the performance of the remaining tasks was flat, indicating that the algorithm we proposed also owns a better stability and sample efficiency on SAC.

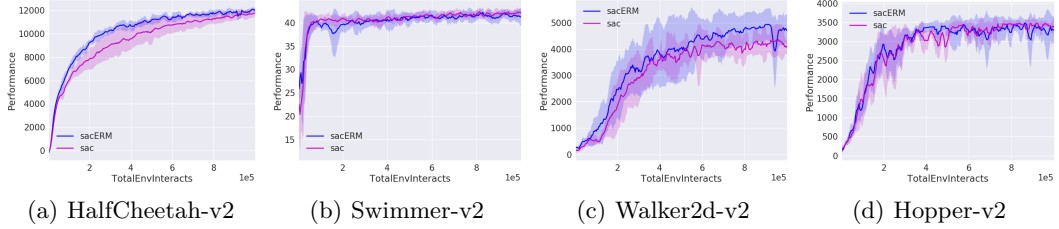

(a) HalfCheetah-v2  (b) Swimmer-v2  (c) Walker2d-v2  (d) Hopper-v2

Figure 6: Performance curves for a selection of domains using SAC and SAC with ERM: SAC (purple), SAC with ERM (blue).

In addition, we summarize the results of the above several algorithms in the HalfCheetah simulation environment. As shown in the sub-graph (a) in Fig. 7. We run agent with each algorithm for 1 million time steps with evaluations every 4000 time steps, where each evaluation reports the average reward over 10 episodes with no exploration noise. Our results are reported over 5 random seeds of the Gym simulator. It can be seen that the performance of SAC with ERM is the best, surpassing state of the art (sac), and its convergence speed is also the fastest, indicating that it has higher sample efficiency compared with other off-policy RL algorithms.

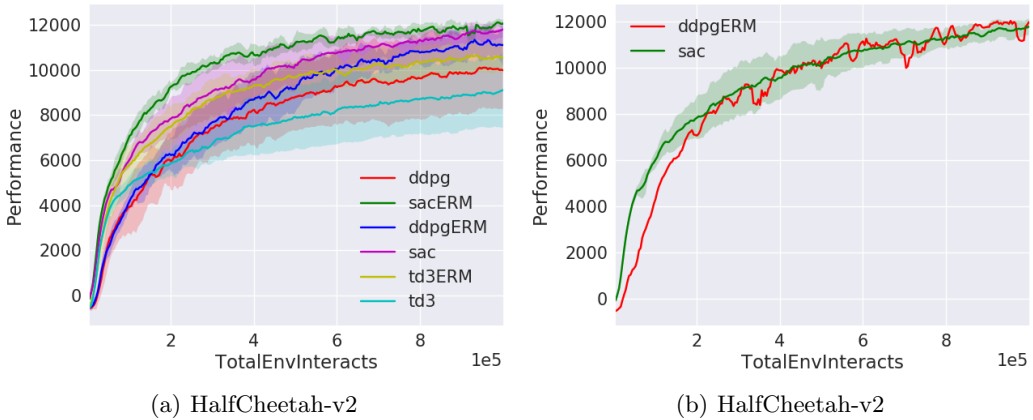

(a) HalfCheetah-v2  (b) HalfCheetah-v2

Figure 7: Performance curves for HalfCheetah-v2 task using different algorithms: (a)DDPG (red), DDPG with ERM (blue), TD3 (cyan), TD3 with ERM (yellow), SAC (purple), SAC with ERM (green) (b) SAC (green), DDPG with ERM (red).

We select the result in DDPG with ERM with the random seed 5 in HalfCheetah-v2 task, we compare it with the average performance of SAC, as shown in the subgraph (b) in Fig. 7. It can be found that although the convergence speed of ddpg with ERM is slower in the early stage, its performance has approached or even exceeded the average performance of SAC in the later stage. This improvement is very incredible for DDPG.

We display the statistical data of all the experimental results in Table 1. The first column indicates the algorithm or policy, among which random indicates that the agent uses a random policy to interact with the environment. The numbers in the table represent the average cumulative rewards obtained by the corresponding algorithm or policy in the environment in a episode with 4000 steps. The numbers in bold are the highest performance scores. It can be seen that the SAC with ERM algorithm has the highest score, followed by the DDPG with ERM algorithm, and ERM has the greatest effect on improving the performance of TD3. The statistical average cumulative reward results show that ERM is very helpful to improve the performance of DDPG, TD3 and SAC.

Table 1: Mean value of the total reward of agent in different tasks

| Environment | HalfCheetah-v2 | Swimmer-v2 | Walker2d-v2 | Hopper-v2 |
|---|---|---|---|---|
| Random | -283±29 | $1 \pm 4$ | $2 \pm 2$ | $19 \pm 6$ |
| DDPG | 7790±2058 | $84 \pm 26$ | $920 \pm 550$ | $1313 \pm 867$ |
| DDPGERM | 8415±1161 | $\mathbf{94 \pm 24}$ | $933 \pm 578$ | $1548 \pm 861$ |
| SAC | 9452±984 | $41 \pm 2$ | $3163 \pm 951$ | $2856 \pm 502$ |
| SACERM | $\mathbf{10219 \pm 645}$ | $40 \pm 3$ | $\mathbf{3564 \pm 1216}$ | $\mathbf{2883 \pm 543}$ |
| TD3 | 7240±1455 | $58 \pm 31$ | $2568 \pm 733$ | $1939 \pm 1442$ |
| TD3ERM | 8795±1023 | $48 \pm 13$ | $3305 \pm 966$ | $2553 \pm 852$ |

In all the experiments, the most time-consuming part of the algorithm is calculating the similarity between states. To solve this problem, we accelerate the process by using matrix operation, which expands the dimension of current state $S_c$ to be consistent with the dimension of key states buffer, so we calculate the similarity with the matrix $S_k$ composed of all key states quickly. Therefore, in the case of a small increase in time consumption, the performance of the algorithm is significantly improved, especially on the HalfCheetah task.

## 5 Conclusion

This work divides the states of the agent into key states and non-key states, and analyzes the reasons why the recent generated key states need to be sampled and trained as soos as possible. For the purpose of sampling key transitions more accurately, we introduce two experiences buffers to store the key transitions and non-key transitions respectively, and set a adjustable coefficient to determine the proportion of transitions sampled from two experience buffers. Besides, we analyze the advatages of sampling the key transitions if we make use of linear distribution function from two perspectives: 1.The distribution of transitions sampled recently are more similar to the distribution of latest experience generated, making transitions more satisfie the assumption of independent and identical distribution (iid); 2.The recent generated experience can be used for the training of the Q value network and policy network of the agent more quickly, so as to make up for the lack of the policy in time. Finally, on the basis of AN2N, the combination of ERM method and DDPG, TD3 or SAC algorithm has a very obvious performance improvement on tasks such as HalfCheetah. The performance of the DDPGERM algorithm with some random seeds can even exceed the average performance of SAC, and SACERM has also outperforms the current state of the art.

Acknowledgments

We would like to thank Feng Pan, Weixing Li, Xiaoxue Feng, Yan Gao and many others at Institute of Pattern Recognition and Intelligent System of BIT for insightful discussions and feedback.

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

## A    Experiment Details

The experiment uses a unified parameter configuration for all algorithms, the two hidden layers are composed of 256 nodes and the neural networks use the rectified non-linearity (Glorot et al., 2011) for all hidden layers. The activation function of the actor output layer uses a tanh for bounding the actions. Adam (Kingma & Ba, 2014) optimizes the weights of the neural networks with a learning rate of $10^3$ for the both of actor and critic networks, discount factor set to 0.99. For the soft target updates, we set $\tau = 0.005$. At the beginning, the agent uses a random policy to generate $10^4$ transitions for the initial training of the network, the proportion of sampling key transitions is fixed by $\min(Prt_{AN2N}, K_t)$, where $Prt_{AN2N} = 0.4 - 0.2 * \frac{t}{totaltime}$, $K_t = 50 * \frac{t}{totaltime}$. We train the networks with minibatch sizes of 100 for all of the tasks, and use a replay buffer size of $10^6$. For DDPG or TD3 with ERM, we set two kind noise $\mathcal{N}_{\text{big}} = 0.4$ and $\mathcal{N}_{\text{small}} = 0.05$, For SAC with ERM, different from the policy generation method of DDPG and TD3, SAC generates the mean and variance of the policy, and then sample a specific action. Therefore, we set the expansion and reduction coefficients of the variance to 1.5 and 0.5.

We test our algorithm in the Mujoco task environment. The state dimension, observation dimension, and action dimension of the tasks are shown in Table 2, and the detailed description of the tasks are shown in Table 3.

Table 2: Dimensionality of the MuJoCo tasks environment: the dimensionality of the underlying physics model dim(s), number of action dimensions dim(a) and observation dimensions dim(o).

| Task name | HalfCheetah-v2 | Swimmer-v2 | Walker2d-v2 | Hopper-v2 |
|-----------|----------------|------------|-------------|-----------|
| dim(s)    | 18             | 27         | 18          | 14        |
| dim(a)    | 6              | 8          | 6           | 4         |
| dim(o)    | 17             | 111        | 41          | 14        |

Table 3: The description of the Mujoco tasks environment

| Task name | Brief description |
|-----------|-------------------|
| HalfCheetah-v2 | The agent should move forward as quickly as possible with a cheetah like body that is constrained to the plane (Wawrzyński & Tanwani, 2013). |
| Swimmer-v2 | This task involves a 3-link swimming robot in a viscous fluid, where the goal is to make it swim forward as fast as possible (Coulom, 2002). |
| Walker2d-v2 | The agent's goal is making a two-dimensional bipedal robot walk forward as fast as possible (Erez et al., 2011). |
| Hopper-v2 | The agent's goal is making a two-dimensional one-legged robot hop forward as fast as possible (Erez et al., 2011). |

## B    Algorithm

Appendix B includes two parts: the first part is the overall structure diagram 8 of the Experience Replay More (ERM) algorithm. The core of the entire algorithm is to classify and store the data stream, and sample the experience with a linear distribution function. The second part is the pseudo code of the algorithm: algorithm pseudo code of DDPG with ERM in Algorithm 2, algorithm pseudo code of TD3 with ERM in Algorithm 3 and algorithm pseudo code of SAC with ERM in Algorithm 4.

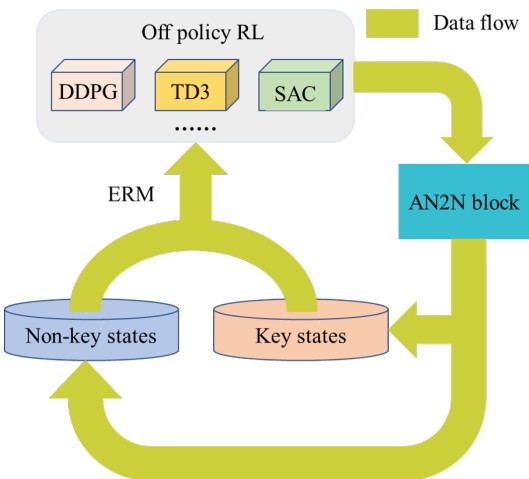

Figure 8: The overall structure diagram of the Experience Replay More (ERM) algorithm.

---

**Algorithm 2: DDPG with ERM**

---

Input: Sampling ratio $Prt_{AN2N}$, $K_t$, Replay buffer $R_{non-key}$, $R_{key}$, batch size $bs_1$, $bs_2$, $bs_{sum}$ and AN2N parameters

Randomly initialize critic network $Q(s, a|\theta^Q)$ and actor $\mu(s|\theta^\mu)$ with weights $\theta^Q$ and $\theta^\mu$

Initialize target network $Q'$ and $\mu'$ with weights $\theta^{Q'} \leftarrow \theta^Q$, $\theta^{\mu'} \leftarrow \theta^\mu$

for episode $e \in \{1,...,M\}$ do

    Initialize a random process $\mathcal{N}$ for action exploration

    Receive initial observation state $s_1$

    for $t \in \{1,...,T\}$ do

        Execute AN2N action $a_t$ and observe reward $r_t$ and observe new state $s_{t+1}$

        if (AN2N exploring more) then

            | Store key transitio $(s_t, a_t, r_t, s_{t+1})$ in $R_{key}$

        else

            | Store transition $(s_t, a_t, r_t, s_{t+1})$ in $R_{non-key}$

        end

        $bs_1 = \min(Prt_{AN2N}, K_t)$

        $bs_2 = bs_{sum} - bs_1$

        Sample $bs_1$ and $bs_2$ transitions with linear distribution in $R_{key}$ and $R_{non-key}$ for training

        if t mod u then

            Sample a random minibatch of $N$ transitions $(s_i, a_i, r_i, s_{i+1})$ from $R$

            Set $y_i = r_i + \gamma Q'(s_{i+1}, \mu'(s_{i+1}|\theta^{\mu'})|\theta^{Q'})$

            Update critic by minimizing the loss:$L = \frac{1}{N}\sum_i(y_i - Q(s_i, a_i|\theta^Q))^2$

            Update the actor policy using the sampled policy gradient:

$$\nabla_{\theta^\mu} J \approx \frac{1}{N}\sum_i \nabla_a Q(s, a|\theta^Q)|_{s=s_i, a=\mu(s_i)} \nabla_{\theta^\mu}\mu(s|\theta^\mu)|_{s_i}$$

            Update the target networks:

$$\theta^{Q'} \leftarrow \tau\theta^Q + (1-\tau)\theta^{Q'}$$
$$\theta^{\mu'} \leftarrow \tau\theta^\mu + (1-\tau)\theta^{Q^{\mu'}}$$

        end

    end

end

---

---

Algorithm 3: TD3 with ERM

---

Input: Sampling ratio $Prt_{AN2N}$, $K_t$, Replay buffer $R_{non-key}$, $R_{key}$, batch size $bs_1$, $bs_2$,
$\quad\quad bs_{sum}$ and AN2N parameters

Randomly initialize critic networks $Q(s,a|\theta_1^Q)$, $Q(s,a|\theta_2^Q)$, and actor $\mu(s|\theta^\mu)$ with
$\quad$ weights $\theta_1^Q$, $\theta_2^Q$ and $\theta^\mu$

Initialize target network $Q'$ and $\mu'$ with weights $\theta_1^{Q'} \leftarrow \theta_1^Q$, $\theta_2^{Q'} \leftarrow \theta_2^Q$, $\theta^{\mu'} \leftarrow \theta^\mu$

for episode $e \in \{1,...,M\}$ do
$\quad$ Initialize a random process $\mathcal{N}$ for action exploration
$\quad$ Receive initial observation state $s_1$
$\quad$ for $t \in \{1,...,T\}$ do
$\quad\quad$ Execute AN2N action $a_t$ and observe reward $r_t$ and observe new state $s_{t+1}$
$\quad\quad$ if (AN2N exploring more) then
$\quad\quad\quad$ | Store key transitio $(s_t, a_t, r_t, s_{t+1})$ in $R_{key}$
$\quad\quad$ else
$\quad\quad\quad$ | Store transition $(s_t, a_t, r_t, s_{t+1})$ in $R_{non-key}$
$\quad\quad$ end
$\quad\quad$ $bs_1 = \min(Prt_{AN2N}, K_t)$
$\quad\quad$ $bs_2 = bs_{sum} - bs_1$
$\quad\quad$ Sample $bs_1$ and $bs_2$ transitions with linear distribution in $R_{key}$ and $R_{non-key}$ for
$\quad\quad\quad$ training
$\quad\quad$ if t mod u then
$\quad\quad\quad$ Sample a random minibatch of $N$ transitions $(s_i, a_i, r_i, s_{i+1})$ from $R$
$\quad\quad\quad$ Get $\tilde{a} \leftarrow \mu(s|\theta^{\mu'}) + \epsilon, \quad \epsilon \sim \text{clip}(\mathcal{N}(0,\tilde{\sigma}), -c, c)$
$\quad\quad\quad$ Set $y_i = r_i + \gamma \min_{j=1,2} Q'(s_{i+1}, \mu'(s_{i+1}|\tilde{a})|\theta_j^{Q'})$
$\quad\quad\quad$ Update critic by minimizing the loss:$L = \min_{\theta_j} \frac{1}{N} \sum_i (y_i - Q(s_i, a_i|\theta_j^Q))^2$
$\quad\quad\quad$ if t mod $(d \times u)$ then
$\quad\quad\quad\quad$ Update the actor policy $\mu(s|\theta^\mu)$ using the sampled policy gradient:

$$\nabla_{\theta^\mu} J \approx \frac{1}{N} \sum_i \nabla_a Q(s,a|\theta_1^Q)|_{s=s_i, a=\mu(s_i)} \nabla_{\theta^\mu} \mu(s|\theta^\mu)|s_i$$

$\quad\quad\quad\quad$ Update the target networks:

$$\theta_j^{Q'} \leftarrow \tau\theta_j^Q + (1-\tau)\theta_j^{Q'}$$
$$\theta^{\mu'} \leftarrow \tau\theta^\mu + (1-\tau)\theta^{Q^{\mu'}}$$

$\quad\quad\quad$ end
$\quad\quad$ end
$\quad$ end
end

---

---

Algorithm 4: SAC with ERM

---

Input: Sampling ratio $Prt_{AN2N}$, $K_t$, Replay buffer $R_{non-key}$, $R_{key}$, batch size $bs_1$, $bs_2$,
$\quad\quad bs_{sum}$ and AN2N parameters, Temperature parameter $\alpha$

Randomly initialize critic networks $Q(s,a|\theta_1^Q)$, $Q(s,a|\theta_2^Q)$, and actor $\mu(s|\theta^\mu)$ with
weights $\theta_1^Q$, $\theta_2^Q$ and $\theta^\mu$

Initialize target network $Q'$ and $\mu'$ with weights $\theta_1^{Q'} \leftarrow \theta_1^Q$, $\theta_2^{Q'} \leftarrow \theta_2^Q$

for episode $e \in \{1,...,\text{M}\}$ do
$\quad$ Initialize a random process $\mathcal{N}$ for action exploration
$\quad$ Receive initial observation state $s_1$
$\quad$ for $t \in \{1,...,\text{T}\}$ do
$\quad\quad$ Execute AN2N action $a_t$ and observe reward $r_t$ and observe new state $s_{t+1}$
$\quad\quad$ if (AN2N exploring more) then
$\quad\quad\quad$ Store key transitio $(s_t, a_t, r_t, s_{t+1})$ in $R_{key}$
$\quad\quad$ else
$\quad\quad\quad$ Store transition $(s_t, a_t, r_t, s_{t+1})$ in $R_{non-key}$
$\quad\quad$ end
$\quad\quad$ $bs_1 = \min(Prt_{AN2N}, K_t)$
$\quad\quad$ $bs_2 = bs_{sum} - bs_1$
$\quad\quad$ Sample $bs_1$ and $bs_2$ transitions with linear distribution in $R_{key}$ and $R_{non-key}$ for
$\quad\quad$ $\quad$ training
$\quad\quad$ if t mod u then
$\quad\quad\quad$ Sample a random minibatch of $N$ transitions $(s_i, a_i, r_i, s_{i+1})$ from $R$
$\quad\quad\quad$ Set $y_i = r_i + \gamma(\min_{j=1,2} Q'(s_{i+1}, \mu(s_{i+1}|\theta^\mu)|\theta_j^{Q'}) - \alpha \log \mu(s_{t+1}|\theta^\mu))$
$\quad\quad\quad$ Update critic (soft Q-function) by minimizing the loss:
$$L = \frac{1}{N} \sum_i \sum_j (y_i - Q(s_i, a_i|\theta_j^Q))^2$$
$\quad\quad\quad$ Update the actor policy using the sampled policy gradient:
$$\nabla_{\theta^\mu} J \approx \frac{1}{N} \sum_i ((\nabla_a Q(s,a|\theta^Q)|_{s=s_i, a=\mu(s_i)} \nabla_{\theta^\mu} \mu(s|\theta^\mu)|s_i$$
$$-\nabla_a \log \mu(s_t|\theta^\mu)) - \nabla_{\theta^\mu} \log \mu(s_t|\theta^\mu)|s_i)$$
$\quad\quad\quad$ Update the target networks:
$$\theta_j^{Q'} \leftarrow \tau\theta_j^Q + (1-\tau)\theta_j^{Q'}$$
$\quad\quad$ end
$\quad$ end
end

---

