# OpenReview forum: "Experience Replay More When It's a Key Transition in Deep Reinforcement Learning"
_ICLR.cc/2022/Conference — ICLR 2022 Submitted_

### Official Review · Reviewer_Jn1b · 2021-10-29

**Correctness:** 1
**Technical Novelty And Significance:** 1
**Empirical Novelty And Significance:** 1
**Recommendation:** 3
**Confidence:** 4

**Main Review:**

- The paper is very difficult to read as it contains many typos and fragmented sentences. This problem is so acute that some paragraphs are almost incomprehensible.
- It is also very unclear how the similarity between states is calculated. For instance in paragraph 3.1.1 this is referred to as a key step to define what transitions are considered ‘key’, but little details are given. This leaving aside that s_c and s_d are not defined in formal terms.
- A clear downside of the paper is that only a single domain is used (DM Control suite) and even worse ERM was tested in only 4 environments, thus questioning the generality of the method.
- Figure 4 highlights the results of DDPG with and without ERM. The only environment where there seems to be a small advantage of DDPG+ERM is HalfCheetah, although the error of the baseline is overlapping. So I strongly disagree with the statements of the last paragraph of section 4.1. Also, visually there is no sign of better data efficiency, and if the authors want to make a claim in this direction they should provide more quantitative measures (e.g. area under the curve)
- Figure 6 highlights the results of SAC with and without ERM. Again, the authors claim that SAC+ERM is better than SAC in half of the task, but this conclusion is not supported by the plots.
- Figure 7b is particularly weird as, in deep reinforcement learning, it is very unscientific to put draw any conclusions based on a single seed.
- Table 1 reports some numbers in bold even though the variance is so high that determining what is the best method is a bit arbitrary. Authors should perform proper statistical analysis before highlighting scores in bold.
- Some baselines are completely missing from the experiments, e.g. baseline+prioritise replay (Schaul et al. 2016)


**Summary Of The Paper:**

The paper proposes a new method (ERM) to bias the decision over which transitions should be replayed more often. In particular, using ERM, some states are deemed as key and they are replayed more often (following also a recency bias). This new method is then tested in 4 environments on the DeepMind control suite by using 3 different RL algorithms (DDPG, SAC, TD3).


**Summary Of The Review:**

None of the claims in the abstract, introduction and conclusion are actually supported by the results. Also the paper is not well written, with lots of typos and fragmented sentences.

---

> ### Author Response · Authors · 2021-11-21
> **We have uploaded the revised paper and corrected the existing spelling and grammar problems. For the questions of the reviewers, we give answers based on the revised paper, published papers and our additional experiments.**
>
> 1. We have revised the spelling, grammatical, and clarity issues existing in the original paper and uploaded the new version of the paper. If it is convenient, you could read the places you are confused about the paper again.
>
> 2. The reviewer thinks that there are a few test environments: ERM is a general module, which is essential to verify the universality of the algorithm, so it needs to be combined with other off-policy RL algorithms to see if there is any performance improvement. Three representative algorithms are selected in the paper. For the test environment, we have selected the four environments tested in DDPG, TD3, and SAC, while there are only five test environments in SAC papers，even though we have done three experiments for each environment (a total of 3 * 4 groups of experiments).
>
> 3. About performance improvement: in Halfcheetah environment, the improvement of (DDPG, TD3, SAC) with ERM has exceeded the improvement effect of many other algorithms (for example, TD3 is an improved algorithm of DDPG, and in our reproduction results, it is found that the performance of TD3 in Halfcheetah and Swimmer is even worse than that of DDPG). Moreover, it can be seen from table 1 that not only the algorithms combined with ERM perform best in all tasks, but also the performance of each algorithm is improved in more than half (3 / 4) of the tasks after it is combined with ERM.
>
> 4. To supplement the background, in reinforcement learning, if algorithm a needs less interaction with the environment when algorithm A and algorithm B achieve the same performance, then we think algorithm A has higher sample efficiency. Therefore, I wonder why you can't see the performance improvement of SACERM in Figure 6. In subgraph (a) and subgraph (c), the blue curve (SACERM ) is generally above the purple curve (SAC). At the same time, in subgraph (a), we can see that SACERM  not only has better performance but also has significantly faster convergence speed. Specific numerical statistics on the performance of different algorithms in different test environments are also recorded in Table 1.
>
> 5. As for figure (7b), we draw this figure separately because the performance of the SAC algorithm is much higher than DDPG and TD3 in the Halfcheetah task. Therefore, when we found that the performance of DDPG with ERM under a specific random seed (only the best one in five experiments) exceeded the average performance of SAC, we think this result is worth showing in the paper.
>
> 6. The stability of deep reinforcement learning algorithms is low, which has long been a consensus in this field. In Table 1, the addition of ERM has the effect of reducing variance in most test environments, except that it increases variance within an acceptable range after combining with sac. From all the experiments done, the addition of ERM plays a role in reducing variance in most test environments, and the role is most obvious in the Halfcheetah environment.
>
> 7. The reason why PER (prioritized experience replay) is not selected as the baseline is that we did some research before selecting the baseline and found that the combined effect of PER with TD3, SAC, and other algorithms is not ideal. For details, see the following papers (1.https://arxiv.org/pdf/2109.11767.pdf 2.https://arxiv.org/pdf/1906.04009.pdf 3.https://arxiv.org/pdf/2006.13169.pdf). Therefore, PER is excluded.

---

### Official Review · Reviewer_Hemq · 2021-10-31

**Correctness:** 1
**Technical Novelty And Significance:** 1
**Empirical Novelty And Significance:** 2
**Recommendation:** 1
**Confidence:** 4

**Main Review:**

Experience replay is a really important component of many RL algorithms right now. The importance of the distribution of replay states has been convincingly demonstrated in prior work. So the topic of this paper is definitely relevant to the ICLR audience.

One of my big concerns is that the paper is poorly written. The grammar, spelling, and word choice are all sufficiently flaws to, at times, obscure meaning. Even beyond that, the substance of the paper is not communicated clearly. While I understand that the idea of the paper is to more frequently replay states that are more important in some way, the paper does not communicate how those states are identified or what makes them more important. While the paper does a reasonable job of motivating prioritized replay in general, it does not clearly motivate or justify its specific approach.

The biggest clarity problem comes from the fact that ERM relies heavily on the AN2N algorithm, which has not been peer-reviewed and has only been available on arXiv for literally a few days at time of reviewing. Given this, the AN2N algorithm needs to be far more precisely described and justified. From the text of this paper, I have no idea how AN2N works or why I should believe that it is a good base to build on. This is critical, because the ERM algorithm makes its decisions based on the output of the AN2N algorithm. Without a clear description and justification of AN2N, the description and justification of ERM is also incomplete. Even if the AN2N paper were more clearly written and its hypotheses clearly evaluated (from a brief skim this does not seem to be the case), it is important that *this* paper be sufficiently self-contained that the reader can understand the significance of the contributions.

Here is a non-exhausted list of some other clarity concerns I encountered:
- Why is Hafner et al. the citation for model-based RL? MBRL predates 2019 by a long shot.
- I don't think it is correct to claim that off-policy model-free learning is categorically more sample efficient than on-policy learning. There are things you can do with off-policy learning that you can't really (justifiable) do with on-policy learning, like experience replay, that can improve sample efficiency. However, the same is true in the reverse; for instance eligibility traces are far more sensible in the on-policy case.
- Page 2: I didn't understand the logical step between discussing (prioritized) experience replay and meta-RL.
- Page 2: I don't understand what the footnotes are meant to communicate.
- Page 3/4: I don't understand Figure 1.
  - Isn't HalfCheetah continuous in state? So what do s0, s1, etc. represent?
  - Each line on the plot is associated with a particular Q-value (I think??), so what does the y-axis represent? The caption says that it is "the specific value of the state."
  - The caption says that "the difference is obvious at states s1, s8, and s12." I don't know what I'm supposed to be comparing in order to see a difference. And I don't see why those three states are special compared to any other states.
  - The main text concludes that "the state is also changing." I don't know what "the state"  refers to here, or what change is being discussed.

My other major concern is that the conclusions are far too strong for the experimental evidence presented. Only five independent trials are run for each algorithm and the error bars are quite large. I couldn't find the explanation of what the error bars actually represent (maybe I missed it), but assuming that it is something like the sample standard deviation the differences in performance are well within the noise in all but a couple of experiments. It is true that across all of the experiments there is a tendency for the ERM curve to be slightly above the non-ERM curve, which lends some credibility to the hypothesis that ERM is an improvement in at least some of these experiments, but even then, the improvement is *very* modest.

Furthermore, the empirical results do not test the central hypotheses of the paper. The paper claims that ERM's (modest) success is a result of specific effects, namely replaying more important/relevant states. It's important to evaluate this hypothesis, for instance by performing ERM but without some of its key features (e.g. without prioritization of recent experience or without the prioritization of "key" states). I also note that the comparison is always between ERM and the base RL algorithm. Since ERM is built directly on top of AN2N, it seems critical to include results using AN2N without the additional enhancements of ERM in order to understand the significance of the contributions of both algorithms.

**Summary Of The Paper:**

The paper presents Experience Replay More, an algorithm that prioritizes certain transitions in a replay buffer for replay. The approach is combined with multiple base RL algorithms in four MuJoCo domains.

**Summary Of The Review:**

While investigating an important problem, the paper is poorly written, the main contribution is not adequately described or evaluated, and the empirical findings that are presented to not convincingly support the conclusions. I recommend that this paper be rejected.

---

> ### Author Response · Authors · 2021-11-21
> **We have uploaded the revised paper and corrected the existing spelling and grammar problems. For the questions of the reviewers, we give answers based on the revised paper and our additional experiments.**
>
> 1. The main idea of the paper is to prioritize sampling (The later the transition is generated, the greater the probability of being sampled) the recently generated "key" states. "Key" states are the states where the agent performs poorly, measured by the return value.
>
> 2. In the newly submitted paper, some expressions in section 3 have been revised again. If it is convenient for you to reread section 3, many doubts will disappear. For example, the caption in Figure 1 describes the meaning of abscissa and ordinate, S0, S1, etc of abscissa represent different parts of the agent, and the ordinate represents the information collected by different parts you can think it is the attitude information of the agent. The figure on the left of Figure 1 collects the state information of the agent's policy not converging (it can be found that the policy is not converging at this time by the Q value marked in the legend, and the maximum Q value is much smaller than the Q value in the figure on the right). The figure on the right shows the state information of different parts of the agent collected when the policy begins to converge. Therefore, the state information of the left figure is different from that of the right figure， the main differences are in S1, S8, S12, and S14. Similarly, in Section 3, according to figure 1, we analyze that the state of the agent in each stage is different. The subset of states that the agent enters after the policy converges is different from the set of states when the policy has not yet begun to converge. If the agent will hardly encounter these states again, then training the Q-value network and policy network with these experience will gradually update the learned network weight. Therefore, we believe that the training times of sampling can be reduced for the states that the agent will not generate or have a small probability. That's why we need to prioritize sampling the recently generated "key" states.
>
> 3. In the Halfcheetah task, we have done some control experiments between ERM and AN2N. The combination of AN2N and TD3 has no performance improvement effect, while the combination of ERM and TD3 has achieved the most obvious performance improvement. From other experiments we have done, it is found that the convergence speed and performance of SAC+ ERM are better than SAC + AN2N, while the effect of DDPG + ERM is slightly better than DDPG + AN2N. This part is indeed not shown in the paper. This part of the experiment will be uploaded in the form of an appendix in the future.
>
> 4. About the extent of performance improvement, first of all, the addition of ERM has significantly improved DDPG, TD3, and SAC in Half cheetah, which is close to or even higher than that of many other improved algorithms. Secondly, ERM is a general module, which can improve not only one algorithm but also many off-policy RL algorithms. As can be seen from Table 1, the performance improvement brought by ERM to DDPG, TD3, SAC, and other algorithms is universal.
>
> 5. Here is just a brief list of the work of model-based RL which indicates that the work related to MBRL needs to establish a world model.
>
> 6. I agree with you that off-policy model-free learning is not categorically more sample efficient than on-policy learning. but it is true in most tasks.

---

> > ### Comment · Reviewer_Hemq · 2021-11-24
> > **Thank you for the response**
> >
> > Thanks to the authors for the response and for the effort in revising the paper. I have reviewed the revised paper, comparing it to the original version.
> >
> > Unfortunately, I find that my most serious concerns are not sufficiently addressed:
> > - I can see that the authors have made an effort to improve the writing but overall the paper remains unclear both in wording and in substance. For example, Section 3 is marginally improved and, with the aid of the authors' comment, I can make a better guess at what it is mean to communicate (that the distribution of visited states is changing over time, I think?). Nevertheless, the section in the paper itself is still very unclear and I find it unlikely that a general reader will be able to interpret these findings or determine their significance.
> > - The paper is still not self-contained. It still relies heavily on AN2N, which has not been peer reviewed, and which is not adequately described in this paper for a reader to understand what algorithm is being proposed/evaluated.
> > - The paper is still missing critical elements of the empirical evaluation, specifically evidence of the main claim that the replay distribution of ERM avoids specific problems of other replay methods and results in better performance. The authors claim to have this evidence but it is either not provided at all (ablation studies with AN2N), or is contained within separate unpublished papers (comparisons to relevant baselines). These results belong in this paper, with prominent emphasis. Without these results the main scientific claims of the paper are unsupported.
> > - The conclusions are still too strong for the available evidence, even in the authors' response which still makes claims of "universal improvement". With low sample sizes and high variances, we simply cannot conclude that ERM yields an improvement in these domains. Even in the case of HalfCheetah, where the authors rightly claim that the performance difference is largest, the variances are also correspondingly large. I don't see clear evidence of a "significant" difference in any of these experiments and the authors have not provided a statistical basis for that assertion. Even if the improvement were significant in HalfCheetah, an idea that improves performance in one problem is at best a problem-specific heuristic, not a general principle. The paper makes broad claims about the failure modes of experience replay and the relative advantages of ERM over other strategies. A modest improvement in a single problem over the most basic baseline does not adequately evaluate those claims.

---

> > > ### Author Response · Authors · 2021-11-25
> > > **Thank you for the response**
> > >
> > > Thank you for your reply. During this period, I have reviewed my paper again. I will improve from the following aspects:
> > > 1. Revise the paper again;
> > >
> > > 2. Add baseline (PER) for comparison
> > >
> > > 3. Add task scenarios, such as Atari and humanoid
> > >
> > > 4. Ablation Experiment
> > >
> > > 5. Some new ideas may be added to increase the improvement of performance. (with regard to the extent of performance improvement, I think the improvement brought by ERM is commendable, especially for td3. However, I should do more experiments to increase my persuasion)
> > >
> > > In addition to the above five points, I wonder if you have anything to add here?  And this revision should not be completed within this deadline.
> > >
> > > Finally, I would like to thank reviewers again for your valuable comments

---

### Official Review · Reviewer_VS76 · 2021-10-31

**Correctness:** 3
**Technical Novelty And Significance:** 2
**Empirical Novelty And Significance:** 2
**Recommendation:** 3
**Confidence:** 5

**Main Review:**

Strengths:

* The paper motivates why the "key" states should be sampled more frequently.

* The proposed sampling scheme is simple to implement, and can be incorporated into many existing offline RL algorithms.

Weaknesses:

* This paper proposes a different sampling scheme than uniform sampling. However, in the experiments, no other sampling schemes are compared at all. The only baseline is uniform sampling. Even though the paper mentioned prioritized experience replay in the Introduction, it is not considered a baseline in the paper.
* The proposed sampling scheme (ERM) is built upon AN2N (Add Noise to Noise). Therefore, there are at least two variants compared to normal sampling schemes: (1) add more noise to actions following AN2N during exploration (2) sample key states and recent states more frequently. In the experiments, we only see the comparison of normal RL algorithms and RL+ERM (for example, DDPG vs DDPG+ERM, SAC vs SAC+ERM). It is unclear whether the performance improvement (if any) comes from AN2N or AN2N + ERM. To answer this, the authors should include the learning curves of DDGP+AN2N, TD3+AN2N, SAC+AN2N.
* From Figure 4,5,6, we can see the benefits of using ERM are minor. The proposed sampling scheme does not lead to better performance in most cases.
* The paper lacks the discussions on how sensitive the proposed sampling scheme is to the extra hyperparameters that it introduces. For example, does $Prt_{AN2N}$ and $K_t$ need to be tuned a lot when the environments change? The paper should include some ablation studies to show how the performance changes as one changes these hyperparameters.
* The most challenging task in the paper is `Walker2D`. How about other more difficult benchmarking tasks such as Humanoid?
* The writing of the paper needs to be improved substantially. Grammar issues should at least be fixed.

**Summary Of The Paper:**

The paper proposes to change the sampling order in the replay buffer by prioritizing sampling the "key" states and most recently generated states. "Key" states are the states where the agent performs poorly, measured by the return value.

**Summary Of The Review:**

The paper needs more baselines, ablations, benchmarking tasks to justify the effectiveness of the proposed sampling scheme.

---

> ### Author Response · Authors · 2021-11-21
> **We have uploaded the revised paper and corrected the existing spelling and grammar problems. For the questions of the reviewers, we give answers based on the published papers and our additional experiments.**
>
> First of all, thank you for your affirmation and valuable suggestions.
>
> 1. Firstly, the off-policy RL algorithm generally uses the uniform sampling method to obtain training transitions from the experience buffer， therefore, the uniform-sampling-based algorithms should be the baselines in the paper. Secondly, the main reason why it is not compared with PER(prioritized experience replay) alone is that before the experiment, we had surveyed and found that the performance of PER combined with TD3 and SAC is general. See following papers (1.https://arxiv.org/pdf/2109.11767.pdf 2.https://arxiv.org/pdf/1906.04009.pdf 3.https://arxiv.org/pdf/2006.13169.pdf) for details. Therefore, we don't think it is necessary to spare space in the paper for the control experiment of PER.
>
> 2. In the Halfcheetah task, we have done some control experiments between ERM and AN2N. The combination of AN2N and TD3 has no performance improvement effect, while the combination of ERM and TD3 has achieved the most obvious performance improvement.  From other experiments we have done, it is found that the convergence speed and performance of SAC+ ERM are better than SAC + AN2N, while the effect of DDPG + ERM is slightly better than DDPG + AN2N.
>
> 3. Neither DDPG, TD3, SAC nor other off-policy RL algorithms have achieved the SOTA effect in all tasks. In our reproduction effect, TD3 is even worse than DDPG in Halfcheetah and swimmer tasks. As can be seen from Table 1, not only do the algorithms combined with ERM perform best in all tasks but also the performance of each algorithm is improved in more than half (3 / 4) of the tasks after it is combined with ERM.
>
> 4. In the appendix, we mentioned that we used the same hyper-parameter (such as $Prt_{AN2N}$, $K_t$) in different tasks. Therefore, these new hyper-parameters are not sensitive to different tasks or environments. In fact, I only adjusted these parameters a few times according to my own experience. Due to the limitation of computing resources and time, I did not make the further fine adjustment.
>
> 5. Since these algorithms (DDPG, TD3, SAC) do not perform well in humanoid tasks, we do not test the performance of these algorithms combined with ERM on humanoid tasks. Later, we will test the performance of (DDPG, TD3, SAC, etc.)+ERM algorithms in humanoid tasks or even more difficult tasks.
>
> 6. We have revised the spelling, grammatical, and clarity issues existing in the original paper and uploaded the new version of the paper. If it is convenient, you can read the places you are confused in the paper again

---

> > ### Comment · Reviewer_VS76 · 2021-11-24
> > **Reponse**
> >
> > I appreciate the authors' reply. However, I don't find my concerns are addressed. Therefore, I am maintaining my score.
> >
> > * There are many prior works on experience replay. But this paper still did not provide any baselines that use a different strategy to make use of the experience replay. Just to name a few: [1], [2], [3] as examples. Even though PER might not help in the mujoco benchmarks, how about other baselines? Also, the improvements on using ERM are minor (e.g., Figure 6) as well while the authors are claiming adding PER also gives minor improvements based on some results reported in prior works. It's hard to judge which "minor improvement" is more minor among ERM and PER. The authors should put these curves in one figure so that we can see the comparison clearly.
> >
> > * On the other hand, PER works well in many environments with discrete action space such as Atari games. Does ERM outperform PER as well on these domains other than mujoco benchmarks?
> >
> > * Regarding the second point in the authors' reply, "In the Halfcheetah task, we have done some control experiments between ERM and AN2N", where are the curves in the paper? I am not able to find them.
> >
> > * From Table 1, I cannot really see that the improvements of using ERM are significant considering the variance of the performance. From the figures (Figure 4, 6), it is clear that the improvements are minor.
> >
> > * I wonder why the authors claim that SAC etc. do not perform well in humanoid tasks. How do the authors define "perform well" in humanoid tasks. Because even the original SAC paper shows descent learning curves on the humanoid environment. [2] also shows that SAC learns well in the Humanoid environment.
> >
> > * I also share the same concern regarding AN2N as Review Hemq. AN2N was just posted on arXiv a few days before the ICLR submission deadline. It is not peer-reviewed and not well-explained in this paper.
> >
> > * I suggest the authors highlight the places in the paper where the writing has been improved.
> >
> >
> >
> > [1]. Novati, Guido, and Petros Koumoutsakos. "Remember and forget for experience replay." International Conference on Machine Learning. PMLR, 2019.
> >
> > [2]. Sinha, Samarth, et al. "Experience replay with likelihood-free importance weights." arXiv preprint arXiv:2006.13169 (2020).
> >
> > [3]. Wang, Che, and Keith Ross. "Boosting soft actor-critic: Emphasizing recent experience without forgetting the past." arXiv preprint arXiv:1906.04009 (2019).

---

> > > ### Comment · Reviewer_Hemq · 2021-11-24
> > > **Comparing paper versions**
> > >
> > > FWIW, at the top of this page you can click "Show Revisions" and then "Compare Revisions" to see highlighted differences.

---

> > > ### Author Response · Authors · 2021-11-25
> > > **Response**
> > >
> > > Thank you for your reply. During this period, I have reviewed my paper again. I will do more experiments according to your opinions, I will improve from the following aspects:
> > > 1. Revise the paper again;
> > >
> > > 2. Add baseline (PER) for comparison
> > >
> > > 3. Add task scenarios, such as Atari and humanoid
> > >
> > > 4. Ablation Experiment
> > >
> > > 5. Some new ideas may be added to increase the improvement of performance. (with regard to the extent of performance improvement, I think the improvement brought by ERM is commendable, especially for td3. However, I should do more experiments to increase my persuasion)
> > >
> > > In addition to the above five points, I wonder if you have anything to add here? This revision should not be completed within this deadline.
> > >
> > > Finally, I would like to thank reviewers again for your valuable comments

---

### Official Review · Reviewer_cBYQ · 2021-11-06

**Correctness:** 2
**Technical Novelty And Significance:** 2
**Empirical Novelty And Significance:** 2
**Recommendation:** 1
**Confidence:** 4

**Main Review:**

## Strengths

1. The idea is simple, and has intuitive reasons for why it could help. With sufficiently large replay buffers and enough changes made to the policy, it de-emphasizes updates on really old/stale transitions that the optimal policy would rarely visit.

## Weaknesses

1. The paper suffers many spelling, grammatical, and clarity issues. It can be difficult to follow, and greatly detracts from the quality of the paper.

2. The paper acknowledges the PER algorithm and criticizes how such prioritization introduces bias which must be corrected by importance sampling. However, the proposed ERM algorithm (or any replay method in general) is similarly biased in that after each update, the data distribution gets increasingly off-policy. Further, despite acknowledgement of this uncorrected off-policyness, many works simply don't correct the data distribution and find that things still tend to work well empirically. Along this note, it seems like PER is a pretty important baseline to compare to in the space of methods which select which transitions to sample from the replay buffer- can the authors comment on why PER was not compared to, and whether the claims of being "state of the art" are reasonable when only comparing to vanilla experience replay?

3. Only 5 runs were used, and across most of the figures, the differences generally don't seem statistically significant. Can the authors comment on whether enough runs were performed, what the shaded regions represent, and whether conclusions can be drawn from the results with reasonable confidence? Similarly, the following line in the abstract is concerning with regards to empirical methodology: "DDPG with ERM can even exceed the average performance of SAC under certain random seeds"- it's not a particularly fair comparison to suggest that one algorithm on specific random seeds can outperform the *average* performance of another algorithm. That said, the Figure showing this comparison (7b) doesn't appear statistically significant and it's unclear whether this conclusion can be drawn from it.

**Summary Of The Paper:**

The paper proposes to set up experience replay such that transitions are emphasized based on whether the AN2N algorithm decides to explore more, and how recent the transition was experienced. It is based on observations that the state distribution can dramatically drift over the course of policy improvement, and the intuition that it might be better to perform updates on states that the current policy is more likely to actually visit (recent transitions). They apply the proposed sampling method, *experience replay more* (ERM), to a variety of replay-based deep RL algorithms and evaluate them empirically.

**Summary Of The Review:**

In light of the above concerns surrounding writing clarity, missing notable baselines, and questionable empirical methodology, I'm recommending strong rejection at this time. I believe a substantial rewrite and significantly more evaluation is needed.

---

> ### Author Response · Authors · 2021-11-21
> **We revised many spelling and grammar problems in the original paper, uploaded a new version of the paper, and answered the concerns of the reviewers one by one.**
>
> 1. For the first question:
> We have revised the spelling, grammatical, and clarity issues existing in the original paper and uploaded the new version of the paper. If it is convenient, you could read the places you are confused in the paper again.
> 2. For the second question:
> The baseline in our paper is selected by referring to the baseline often compared in off-policy RL related papers. TD3, especially the SAC algorithm, still maintains the SOTA level in some mojoco continuous control tasks. The reason why PERis not selected as the baseline is that we did some research before selecting the baseline and found that the combined effect of PER with TD3, SAC, and other algorithms is not ideal. For details, see the following papers (1.https://arxiv.org/pdf/2109.11767.pdf 2.https://arxiv.org/pdf/1906.04009.pdf 3.https://arxiv.org/pdf/2006.13169.pdf). Therefore, PER is excluded.
> 3. For the third question:
> During the period of proposing ERM, we conducted a large number of experiments (more than 20 times) for a single scene (Halfcheetah), and found that ERM always achieved better results and had high repeatability. Therefore, in the subsequent experiments, we ran 5 experiments for each task, which was also consistent with the number of experiments in SAC paper. See section 5.1 in SAC(http://proceedings.mlr.press/v80/haarnoja18b/haarnoja18b.pdf) for details. As for figure (7b), we draw this figure separately because the performance of the SAC algorithm is much higher than DDPG and TD3 in Halfcheetah task. Therefore, when we found that the performance of DDPG with ERM under a specific random seed (only the best one in five experiments) exceeded the average performance of SAC, we think this result is worth showing in the paper.

---

> > ### Comment · Reviewer_cBYQ · 2021-11-24
> > **Thank you for the response**
> >
> > I appreciate the response and some of the clarifications behind choices made in the paper. However, I don't think my concerns were addressed:
> >
> > As Reviewer VS76 pointed out, there are other experience replay extensions beyond PER, and those reasons for not including PER may be domain-specific things. If there are domains where PER leads to substantial improvements, it might highlight issues with the scope of the empirical evaluation done to make the claims in the paper. Beyond that, can you comment on the critique about PER introducing bias? It seems to suggests that ERM (or experience replay in general) does not introduce any, which seems off.
> >
> > Regarding the number of runs used, when you found that it had high repeatability, just how high was this repeatability? Citing that another paper also used 5 runs isn't really justification for the choice- to use this empirical evaluation in support of claims that the proposed algorithm is better, you need to make a statistical argument suggesting how certain you are that the algorithm didn't just get lucky. What statistical significance test was used, and is it compatible with the number of runs provided (e.g., does it rely on central tendency, and are 5 samples enough for that)? This is similarly what makes the comparison between a single run of one algorithm and the average performance of another algorithm seem misleading to report.

---

> > > ### Author Response · Authors · 2021-11-25
> > > **Response**
> > >
> > > Thank you for your reply. During this period, I have reviewed my paper again. I will do more experiments according to your opinions, but I don't think it is necessary to increase the number of experiments. The reason is that if I increase the scene of the task, I also increase the number of experiments in a certain sense, and if I continue to increase the number of experiments of each task, The total number of experiments will be very large. At present, the number of experiments has reached 3 * 4 * 5 = 120 (3:3 algorithms, ddpg + ERM, td3 + ERM, sac + ERM. 4:4 tasks, halfchettah..., 5: each experiment runs 5 times). This does not include the experiments of the control group. I think there are a large number of experiments at present, and we will add new tasks later.
> > >
> > > I will improve from the following aspects:
> > > 1. Revise the paper again;
> > >
> > > 2. Add baseline (PER) for comparison
> > >
> > > 3. Add task scenarios, such as Atari and humanoid
> > >
> > > 4. Ablation Experiment
> > >
> > > 5. Some new ideas may be added to increase the improvement of performance. (with regard to the extent of performance improvement, I think the improvement brought by ERM is commendable, especially for td3. However, I should do more experiments to increase my persuasion)
> > >
> > > In addition to the above five points, I wonder if you have anything to add here? This revision should not be completed within this deadline.

---

> > > > ### Comment · Reviewer_cBYQ · 2021-11-28
> > > > **Re: Response**
> > > >
> > > > I appreciate the aspects in which you plan to improve the paper, but strongly recommend you consider why more runs might be needed. As several have pointed out, many of the results don't look statistically significant. Even if one algorithm's mean is higher across the five runs, there is high variance that it's hard to say if the algorithm just got lucky or not. More runs, along with proper statistical significance testing, provides a way to quantify the argument you're trying to make. If you simply increase the number of environments and algorithms to compare, but still do 5 runs, you might end up with many more results *with the exact same problem*. While it's true that there are a lot of experiments to run, it would be worse if many experiments were run and there were no statistically significant conclusions you can draw from them. I encourage the authors to check out Henderson et al. (2017) and Colas et al. (2018), as they go in more detail as to why 5 runs is generally not sufficient, even if other published works have been getting away with it.

---

> > > > > ### Author Response · Authors · 2021-12-02
> > > > > **Re:Re:response**
> > > > >
> > > > > I've read the two papers you recommended and realized the problem of insufficient runs. In the follow-up work, I will increase the runs in each task to make the experimental results more statistical significance. Thank you again for your patient and detailed reply.

---

### Decision · Program_Chairs · 2022-01-20

**Decision:**

Reject

**Comment:**

The paper proposes an interesting way of prioritizing samples in replay that is compatible with many RL methods. It is evaluated experimentally on different tasks and with different RL algorithms.

The reviewers highly appreciated the revised paper and the detailed replies and discussions.
While this iteration improved the paper substantially, it is still not ready for publication in its current form. In particular:
- The paper is still not self-contained enough
- The reviewers are still not convinced about the statistical significance
- More tasks should be added
- PER needs to be added as a baseline
The authors promised those changes for the final version, but those are so substantive that the paper will need to go thorough another complete review cycle. Hence, we'd like to encourage the authors to re-submit at a different venue.

P.S.: Careful with double-blind submissions, acknowledgements should not be included.